# Optical Properties Investigation of Upconverting K_2_Gd(PO_4_)(WO_4_):20%Yb^3+^,Tm^3+^ Phosphors

**DOI:** 10.3390/ma16031305

**Published:** 2023-02-03

**Authors:** Julija Grigorjevaite, Arturas Katelnikovas

**Affiliations:** Institute of Chemistry, Faculty of Chemistry and Geosciences, Vilnius University, Naugarduko 24, LT-03225 Vilnius, Lithuania

**Keywords:** luminescence, decay curves, upconversion emission, energy transfer, CIE1931 color coordinates

## Abstract

Nowadays, scientists are interested in inorganic luminescence materials that can be excited with UV or NIR radiation and emit in the visible range. Such inorganic materials can be successfully used as luminescent or anti-counterfeiting pigments. In this work, we report the synthesis and optical properties investigation of solely Tm^3+^ doped and Yb^3+^/Tm^3+^ co-doped K_2_Gd(PO_4_)(WO_4_) phosphors. The single-phase samples were prepared using a solid-state reaction method. The Tm^3+^ concentration was changed from 0.5% to 5%. Downshifting and upconversion emission studies were performed under 360 nm and 980 nm excitation, respectively. Yb^3+^ ions were used as sensitizers in the K_2_Gd(PO_4_)(WO_4_) phosphors to transfer the captured energy to Tm^3+^ ions. It turned out that under UV excitation, phosphors emitted in the blue spectral area regardless of the presence or absence of Yb^3+^. However, a very strong deep-red (~800 nm) emission was observed when Yb^3+^ and Tm^3+^-containing samples were excited with a 980 nm wavelength laser. It is interesting that the highest upconversion emission in the UV/Visible range was achieved for 20% Yb^3+^, 0.5% Tm^3+^ doped sample, whereas the sample co-doped with 20% Yb^3+^, 2% Tm^3+^ showed the most intensive UC emission band in the NIR range. The materials were characterized using powder X-ray diffraction and scanning electron microscopy. Optical properties were studied using steady-state and kinetic downshifting and upconversion photoluminescence spectroscopy.

## 1. Introduction

The lanthanides doped inorganic upconversion (UC) materials with excellent optical properties have significant applications in wide fields, for example, temperature sensing [1], solar light conversion [2], optical sensors [3], or security applications [4], bioimaging [5], optical heating [6], optogenetics [7], nanoscopy [8], nanoscale optical writing [9], etc. Inorganic phosphors, compared with organic dyes or quantum dots, have several advantages, including a longer excited state lifetime, sharp emission bandwidths, and cheaper and more environmentally friendly synthesis [4]. UC is a process where at least two low-energy photons produce high-energy photons. Usually, the inorganic upconverting luminescent materials contain two incorporated RE^3+^ ions: one as a sensitizer, typically Yb^3+^, and another as an emitter, for example, Ho^3+^ [10,11], Tm^3+^ [12,13], etc. Yb^3+^ possesses a simple energy level structure, including one ground (^2^F_7/2_) and one excited (^2^F_5/2_) state level. Additionally, the excited state of Yb^3+^ is higher than the metastable energy levels of emitters in UC materials, for example, Er^3+^, Ho^3+^, or Tm^3+^. In this case, the energy of ^2^F_7/2_ → ^2^F_5/2_ transition is capable of exciting other rare-earth (RE^3+^) elements, and UC emitters can release emissions.

The UC process in the phosphors could be achieved through co-doped RE^3+^ ions into a suitable host lattice. The K_2_Gd(PO_4_)(WO_4_), as a novel phosphate-tungstate compound, has been suitable for application in important optical material fields [14]. Tungstate and phosphate could be used as hosts due to their excellent structure and high thermal stability. Thus, in tungstate, the average distance between the luminescent centers is larger, which may cause a reduced concentration quenching of RE^3+^ [15], which indicates that this host lattice has a high tolerance for heavily doped RE^3+^ [16]. Thus, the RE^3+^ concentration could be an important factor for the luminescence properties, and the proper ratio between a sensitizer and an emitter should be chosen.

The purpose of achieving blue light under 980 nm excitation inspired us to synthesize materials co-doped with Yb^3+^-Tm^3+^ because this RE^3+^ pair is the best combination for blue emission because, among the rare-earth ions, Tm^3+^ is one of the most studied RE^3+^ for blue emission based upon upconversion mechanism. In this work, co-doped crystalline K_2_Gd(PO_4_)(WO_4_):20%Yb^3+^,Tm^3+^ phosphors are synthesized by a simple solid-state reaction and analyzed for the first time to the best of our knowledge. Further study of K_2_Gd(PO_4_)(WO_4_) samples exhibits both downshifting and upconversion photoluminescence under 360 nm and 980 nm excitation, respectively. The obtained results of the synthesized materials show great potential for NIR-excited security pigments application.

## 2. Materials and Methods

Two series of K_2_Gd(PO_4_)(WO_4_) (further abbreviated as KGPW) samples were solely doped with Tm^3+^ and co-doped with 20% Yb^3+^ and Tm^3+^. The Tm^3+^ concentration was 0%, 0.5%, 1%, 2%, and 5% with respect to Gd^3+^. All samples were prepared by the solid-state reaction method. The stoichiometric amounts of reagents, namely, Gd_2_O_3_ (99.99% Tailorlux, Münster, Germany), K_2_CO_3_ (99+% Acros Organics, Geel, Belgium), NH_4_H_2_PO_4_ (99% Reachem Slovakia, Petržalka, Slovakia), WO_3_ (99+% Acros Organics), Yb_2_O_3_ (99.99% Alfa Aesar, Haverhill, MA, USA), and Tm_2_O_3_ (99.99% Alfa Aesar) were weighed, poured to an agate mortar, and moistened with some acetone. The moist mixture of the materials was homogenized until all the acetone evaporated. The dry homogenous powders were placed in the porcelain crucibles and sintered for 10 h at 873 K. To obtain single-phase compounds, the sintering procedure was repeated two more times.

The phase purity of the prepared compounds was checked using a Rigaku MiniFlexII diffractometer (Tokyo, Japan). The XRD patterns were collected in the 2θ range of 5° to 80°. IR spectra were obtained within the range of 4000 to 400 cm^−1^ using a Bruker Alpha ATR spectrometer (Ettlingen, Germany) with a resolution of 4 cm^−1^. The samples were also examined using the FE-SEM SU-70 microscope from Hitachi (Tokyo, Japan). Room temperature and temperature-dependent optical properties of the synthesized compounds were investigated using the FLS980 spectrometer from Edinburgh Instruments (Livingston, UK). The used spectrometer settings are given in Appendix A.

The lattice parameters of the synthesized compounds were calculated from the XRD patterns using the Rietveld refinement method. FullProf Suite software (version 2 December 2022) was used for calculations. The background was set as a 24-term Chebychev-type function. A pseudo-Voigt peak shape was used for the peak profiles. The scale factor, instrument zero, unit cell parameters, preferred orientation, atomic coordinates, and the peak shape (*u*, *v*, *w*, *γ*_0_, and *γ*_1_) parameters were also refined. K_2_Ho(PO_4_)(WO_4_) (PDF-4+ (ICDD) 04-015-9304) compound, reported by Terebilenko et al. [17], was used for Rietveld fits.

## 3. Results and Discussion

Overall, two different series of samples were prepared. One contained KGPW doped solely with Tm^3+^, whereas the second one contained samples co-doped with Yb^3+^ and Tm^3+^. The Tm^3+^ concentration in both series varied from 0.5% to 5%. The concentration of Yb^3+^ in the samples was 20%. Tm^3+^-containing samples could be directly excited by UV radiation (~360 nm) (^3^H_6_ → ^1^D_2_ transition (blue upward arrow in Figure 1). The Yb^3+^ co-doped samples, on the other hand, could also be indirectly excited with a 980 nm wavelength laser. Here, the Yb^3+^ ions absorb the laser radiation (^7^F_7/2_ → ^2^F_5/2_ transition) and transfer the captured energy to Tm^3+^, which, after receiving several quants of energy, emits in the UV and visible (Vis) spectral areas. The schematic diagram showing the main Yb^3+^ and Tm^3+^ energy levels involved in the downshifting (DS) and upconversion (UC) processes is depicted in Figure 1. Due to the rather large amount of energy levels, Tm^3+^ can emit in a wide range, i.e., from UV to deep-red and even infrared (IR) [18,19].

The phase purity of all synthesized compounds was checked by recording their powder XRD patterns. The lattice parameters of the synthesized compounds were calculated from the XRD patterns using the Rietveld refinement method. The Rietveld refinement of undoped KGPW, KGPW:5%Tm^3+^, KGPW:20%Yb^3+^, and KGPW:20%Yb^3+^,5%Tm^3+^ specimens are given in Figure 2. The calculated unit cell parameters of undoped KGPW, KGPW:5%Tm^3+^, KGPW:20%Yb^3+^, and KGPW:20%Yb^3+^,5%Tm^3+^ specimens are summarized in Appendix A. The unit cell parameters decrease with increasing Yb^3+^ and Tm^3+^ concentration since both ions are smaller than Gd^3+^. The recorded XRD patterns matched well with the reference pattern, and no additional peaks were observed; therefore, we can conclude that single-phase materials were obtained. K_2_Gd(PO_4_)(WO_4_) is isostructural with the reported K_2_Ho(PO_4_)(WO_4_) compound. The crystal structure of the prepared materials is orthorhombic, and the space group is *Ibca* (#73) [17]. The crystal structure of these compounds is built from isolated PO_4_ and WO_4_ tetrahedral units and K^+^ and Gd^3+^ eight-fold coordinated polyhedra. Considering the same charge and very close ionic radii, we assumed that _VIII_Yb^3+^ (0.985 Å) and _VIII_Tm^3+^ (0.994 Å) replaced _VIII_Gd^3+^ (1.053 Å) ions [20].

The SEM images of KGPW:5%Tm^3+^, KGPW:20%Yb^3+^, and KGPW:20%Yb^3+^,5%Tm^3+^ samples are shown in Appendix A. The SEM images show that samples consist of irregularly shaped and agglomerated particles. No obvious changes in particle shape and size with changing the dopant concentration were observed.

The IR spectra of KGPW and KGPW:20%Yb^3+^ are shown in Appendix A. Both spectra possess several sets of absorption lines in the range of 1100–400 cm^−1^. The three sharp absorption lines at 650–450 cm^−1^ are assigned to the bending vibrations of PO_4_. The strong absorption band ranging from 900 to 700 cm^−1^ is attributed to the stretching vibrations of Mo−O within MoO_4_ tetrahedral units. The strong absorption band at ca. 1075 is ascribed to asymmetric vibrations of PO_4_ tetrahedral units, whereas the band at ca. 935 cm^−1^ is ascribed to symmetric ones [21].

The reflection spectra of undoped KGPW, KGPW:20%Yb^3+^, and KGPW:5%Tm^3+^ are presented in Figure 3. The reflection spectrum of KGPW:20%Yb^3+^,5%Tm^3+^ is identical to the one of KGPW:5%Tm^3+^; therefore, it was not shown. All samples possessed a white body color showing that the samples do not absorb in the VIS range. It also should be mentioned that the reflectance at longer wavelengths is almost 100% showing low defect concentration in the synthesized materials.

The reflection spectra were measured in a 250–750 nm range. The reflection spectra of Tm^3+^ doped samples contain three typical Tm^3+^ absorption lines, i.e., ^3^H_6_ → ^1^D_2_ (ca. 355–370 nm), ^3^H_6_ → ^1^G_4_ (ca. 460–486 nm), and ^3^H_6_ → ^3^F_2,3_ (ca. 655–715 nm) [22]. The broad absorption band in the UV range (around 250–300 nm) could be assigned to the charge transfer from O^2−^ to W^6+^ transition in the host lattice [23].

The excitation (λ_em_ = 450 nm) spectra of KGPW:Tm^3+^ and KGPW:20%Yb^3+^,Tm^3+^ (where the Tm^3+^ concentration is changed from 0.5% to 5%) samples were recorded from 250 to 430 nm and are shown in Figure 4a,c, respectively. The excitation spectra contain one band at 360 nm originating from the typical Tm^3+^ ground state ^3^H_6_ absorption to the excited state ^1^D_2_. In both cases, the highest intensity was achieved with a 5% Tm^3+^ doped sample. Relatively lower excitation intensity in co-doped samples could be explained due to Tm^3+^ → Yb^3+^ energy transfer [24]. The same tendency was observed in emission spectra (λ_ex_ = 360 nm). The highest emission intensity was observed for KGPW:5%Tm^3+^ and KGPW:20%Yb^3+^,5%Tm^3+^ samples (please refer to Figure 4b,d, respectively). The insets in Figure 4b,d show the normalized integrated emission intensity of the prepared samples and reveal that the emission intensity increases with increasing Tm^3+^ concentration and reaches maximum intensity when Tm^3+^ concentration is the highest. There are few sets of emission lines in the down-conversion emission spectra: the intense blue emission at 440–463 nm corresponds to the ^1^D_2_ → ^3^F_4_ transition, whereas much weaker emission lines at 463–485 nm, 650–670 nm, 740–760 nm, and 780–800 nm correspond to the ^1^G_4_ → ^3^H_6_, ^1^G_4_ → ^3^F_4_, ^1^D_2_ → ^3^F_3_, and ^3^H_4_ → ^3^H_6_ transitions, respectively [25]. The most intense emission line observed at 450 nm could be explained due to the directly excited ^1^D_2_ energy level with 360 nm excitation. Upon 360 nm excitation, phosphors doped with Tm^3+^ emit intense blue emission through ^1^D_2_ → ^3^F_4_ transition and that corresponds well with CIE 1931 chromaticity coordinates, depicted in Appendix A. The color coordinates of all the synthesized samples under 360 nm are located in the blue spectral region and are near the perimeter of the CIE 1931 color space diagram. This indicates high color purity. The precise values of color coordinates are tabulated in Appendix A.

The influence of Tm^3+^ concentration on the UC emission intensity was also investigated. The UC emission spectra of KGPW:20%Yb^3+^,Tm^3+^ (where Tm^3+^ concentration is 0.5%, 2%, and 5%) samples under 980 nm laser excitation are given in Figure 5. The observed emission bands in the UV, visible, and near-IR range can be attributed to ^1^D_2_ → ^3^H_6_ (at 355–365 nm), ^1^D_2_ → ^3^F_4_ (at 445–458 nm), ^1^G_4_ → ^3^H_6_ (458–496 nm), ^1^G_4_ → ^3^F_4_ (at 625–670 nm, respectively), and ^3^H_4_ → ^3^H_6_ (at 755–844 nm) transition. The substantial change in UC emission spectra was observed as a function of Tm^3+^ concentration. Interestingly, the intensity of the bands in the UV and visible range do not follow the same trend as the emission band in the NIR range. The 0.5% Tm^3+^ doped sample showed the strongest emission in the UV/Visible range. On the other hand, the 2% Tm^3+^ doped sample yielded the most intensive UC emission band in the NIR range. This sample also showed the strongest overall UC emission, as shown in the inset graph of Figure 5. When Tm^3+^ concentration increases, the Yb^3+^ → Tm^3+^ energy becomes more efficient because the average distance between these ions decreases. However, a further increase in Tm^3+^ concentration leads to a decrease in ^3^H_4_ → ^3^H_6_ emission intensity. Typically, decreasing emission intensity with increasing Tm^3+^ concentration is attributed to the concentration quenching [26]. Furthermore, with increasing Tm^3+^ concentration the probability of energy back transfer from Tm^3+^ to Yb^3+^ increases (^1^G_4_ (Tm^3+^) + ^2^F_7/2_ (Yb^3+^) → ^3^H_5_ (Tm^3+^) + ^2^F_5/2_ (Yb^3+^)), thus, depopulating the ^1^G_4_ (Tm^3+^) level and reducing blue emission [27]. Such energy back transfer also increases the population of ^3^F_4_ (Tm^3+^) level due to non-radiative relaxation from the ^3^H_5_ (Tm^3+^). The populated ^3^F_4_ level can again receive the energy from Yb^3+^ and be excited to ^3^F_2_ level (Tm^3+^) (^2^F_5/2_ (Yb^3+^) + ^3^F_4_ (Tm^3+^) → ^2^F_7/2_ (Yb^3+^) + ^3^F_2_ (Tm^3+^)). Then ^3^F_2_ energy level can populate the ^3^H_4_ level through non-radiative relaxation, thus, increasing the ^3^H_4_ → ^3^H_6_ emission in the NIR region [28].

For a further understanding of the DC process, the PL decay curves for the most intense DC transition ^1^D_2_ → ^3^F_4_ (λ_ex_ = 360 nm, λ_em_ = 450 nm) as a function of Tm^3+^ concentration were recorded. The mono exponential PL decay curves for KGPW:Tm^3+^ and KGPW:20%Yb^3+^,Tm^3+^ samples were obtained, as depicted in Figure 6a,b, respectively. With increasing Tm^3+^ concentration, the PL decay curves get steeper, showing that the effective PL lifetime (*τ_eff_*) values decrease. This, indeed, was confirmed after calculating the *τ_eff_* values [29]:(1)$$\tau_{eff}{}_{}=\frac{\int_{0}^{\infty }I(t)tdt}{\int_{0}^{\infty }I(t)dt}$$

Here, *I(t)* is emission intensity at a given time *t*. The increasing Tm^3+^ concentration leads to decreasing *τ_eff_* values which, in turn, also indicate the decreasing internal quantum efficiency of Tm^3+^.

For a better understanding of the UC process, the PL decay curves of two main Tm^3+^ emission peaks, namely, ^1^G_4_ → ^3^H_6_ (λ_ex_ = 980 nm, λ_em_ = 478 nm) (see Figure 7) and ^3^H_4_ → ^3^H_6_ (λ_ex_ = 980 nm, λ_em_ = 800 nm) (see Figure 8), were measured. The bi-exponential PL decay curves were observed in both cases.

With increasing Tm^3+^ concentration, the UC PL decay curves of the main emission peaks become steeper, suggesting that *τ_eff_* values decrease. The calculated *τ_eff_* values decreased from 201 μs to 107 μs for ^1^G_4_ → ^3^H_6_ transition and from 267 μs to 101 μs for ^3^H_4_ → ^3^H_6_ transition when Tm^3+^ concentration was increased from 0.5% to 5%. The calculated rise time values for ^1^G_4_ → ^3^H_6_ transition decreased from 73 μs to 38 μs when Tm^3+^ concentration was increased from 0.5 % to 5%. Energy transfer from Yb^3+^ to Tm^3+^ increases with increasing Tm^3+^ concentration, which is reflected by a shorter rise time in the KGPW:20% Yb^3+^ with a higher concentration of Tm^3+^. The calculated PL rise and *τ_eff_* values are tabulated in Appendix A.

To evaluate the Tm^3+^ concentration-dependent PL lifetime values of Yb^3+^, the samples were excited with a 980 nm laser, and the PL decay curves for Yb^3+ 2^F_5/2_ → ^2^F_7/2_ transition were recorded. The Yb^3+^ emission was monitored at 1050 nm. The recorded PL decay curves as a function of Tm^3+^ concentration are depicted in Figure 9. As it was expected, the PL lifetime values of the mentioned Yb^3+^ transition drastically decreased from 1312 ± 24 μs to 416 ± 4 μs when Tm^3+^ concentration was increased from 0% to 5%. Such an expected decrease in Yb^3+ 2^F_5/2_ → ^2^F_7/2_ transition PL lifetime with increasing Tm^3+^ concentration is caused by Yb^3+^ → Tm^3+^ energy transfer. The Yb^3+^ → Tm^3+^ energy transfer efficiency (*η_tr_*) was determined from the Yb^3+^ PL lifetime values using this equation [30]:(2)$$\eta_{tr}=\left(1-\frac{\tau_{Yb-Tm}}{\tau_{Yb}}\right)\times 100\%$$

Here, *τ_Yb−Tm_* and *τ_Yb_* are Yb^3+^ PL lifetime values of ^2^F_5/2_ → ^2^F_7/2_ transition in the presence and absence of Tm^3+^, respectively. The *η_tr_* coherently increases from 44% to 68% when changing Tm^3+^ concentration from 0.5% to 5%, respectively. The exact PL lifetime values, together with calculated *η_tr_* values, are given in Figure 9 insets and Appendix A.

Finally, in order to represent an emission color of the synthesized material under different excitation wavelengths, the response of the standard human eye should be considered. For this reason, the color coordinates (in CIE 1931 color space) of the samples, excited with 360 and 980 nm wavelength radiation, were calculated and are given in Figure 10a,b, respectively. Additionally, the exact color coordinate values are given in Appendix A. The color coordinates are invariant upon the Tm^3+^ concentrations but have a slight dependence on the excitation wavelength. The color coordinates of KGPW:20%Yb^3+^,Tm^3+^ samples excited with 980 nm wavelength laser shift upwards if compared to color coordinates obtained for 360 nm excitation. The shift is caused by the fact that the strongest emission lines at 360 nm excitation are at 450 nm (^1^D_2_ → ^3^F_4_ transition), whereas for 980 nm wavelength laser excitation, the ^1^G_4_ → ^3^H_6_ transition at ~475 nm is the strongest one. Color coordinates for samples excited with 360 nm radiation are closer to the perimeter of the CIE 1931 color space diagram showing higher color purity. However, color coordinates slightly shift to the center of the color space diagram if samples are excited with a 980 nm wavelength laser. The shift is caused by relatively strong emission from ^1^G_4_ → ^3^F_4_ transition at ~650 nm if compared to samples excited with 360 nm (please refer to Figure 4d and Figure 5).

## 4. Conclusions

In summary, we have successfully synthesized single-phase K_2_Gd(PO_4_)(WO_4_):Tm^3+^ and K_2_Gd(PO_4_)(WO_4_):20%Yb^3+^,Tm^3+^ powders, where Tm^3+^ concentration varied from 0.5% to 5%. Under 360 nm excitation, there was no concentration quenching (at least up to 5% Tm^3+^) in solely Tm^3+^ doped samples. The highest upconversion emission in the UV/Visible range was achieved with 20%Yb^3+^,0.5%Tm^3+^ concentration. However, the sample co-doped with 20%Yb^3+^,2%Tm^3+^ shows the most intense UC emission band in the NIR range, which could be explained by a more efficient Yb^3+^ → Tm^3+^ energy transfer. Thus, the color coordinates are invariant upon the Tm^3+^ doping concentrations and take place in the blue region.

## Figures and Tables

**Figure 1 materials-16-01305-f001:**
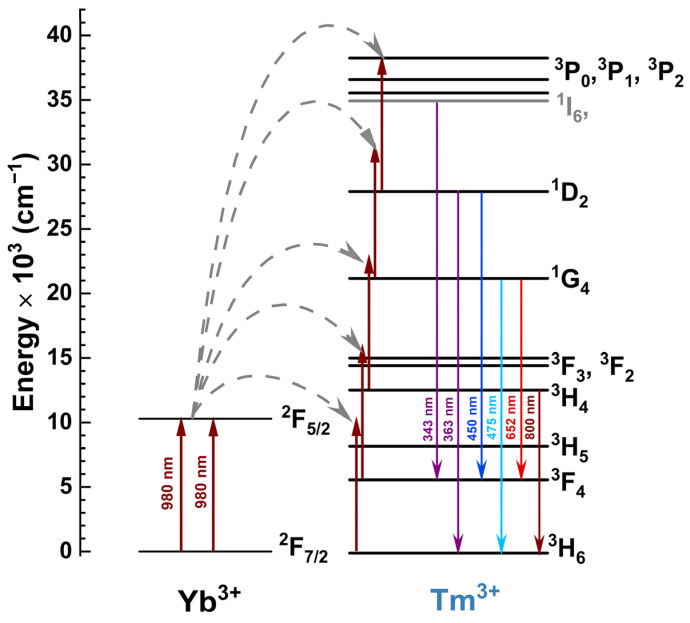
The schematic energy level structure of Yb^3+^ and Tm^3+^.

**Figure 2 materials-16-01305-f002:**
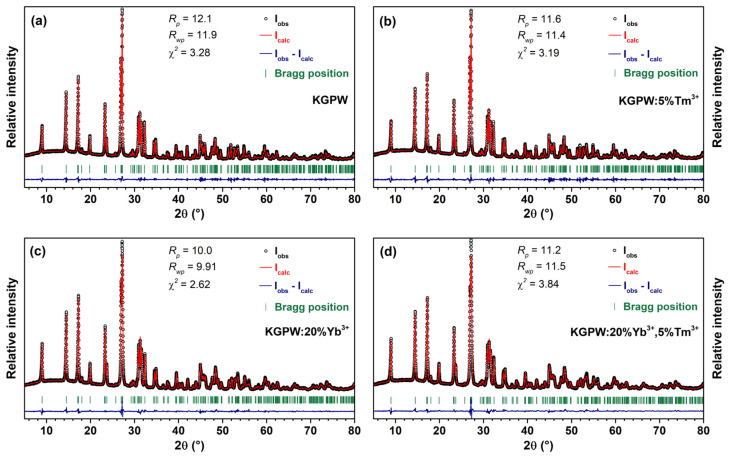
Rietveld refinement of undoped KGPW (**a**), KGPW: 5%Tm^3+^ (**b**), KGPW: 20%Yb^3+^ (**c**), and KGPW: 20%Yb^3+^,5%Tm^3+^ (**d**) XRD patterns.

**Figure 3 materials-16-01305-f003:**
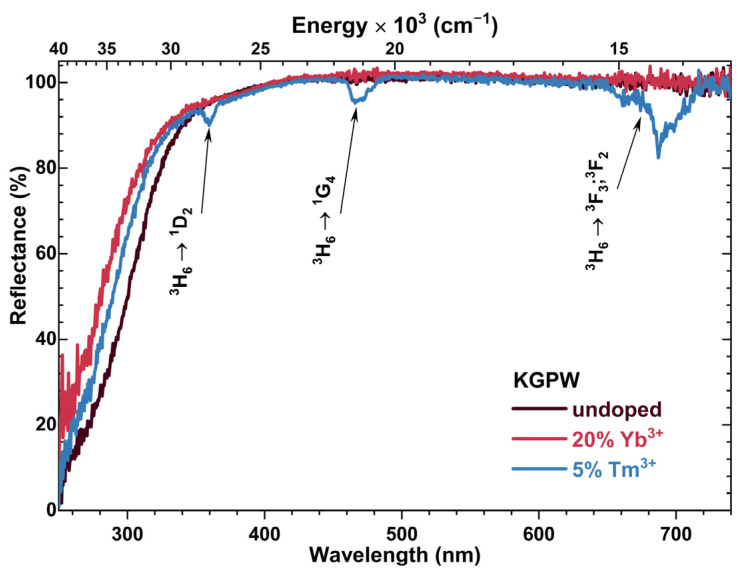
Reflection spectra of undoped KGPW, KGPW: 20%Yb^3+^, and KGPW: 5%Tm^3+^ samples.

**Figure 4 materials-16-01305-f004:**
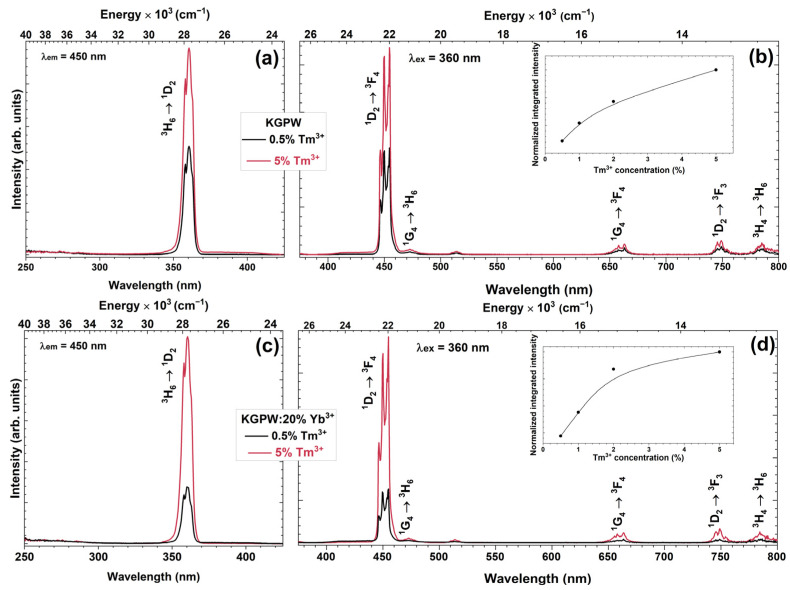
Excitation (λ_em_ = 450 nm) and emission (λ_ex_ = 360 nm) spectra of KGPW: Tm^3+^ (**a**,**b**), and KGPW: 20%Yb^3+^,Tm^3+^ (**c**,**d**), respectively. Both insets show normalized integrated emission of the prepared samples.

**Figure 5 materials-16-01305-f005:**
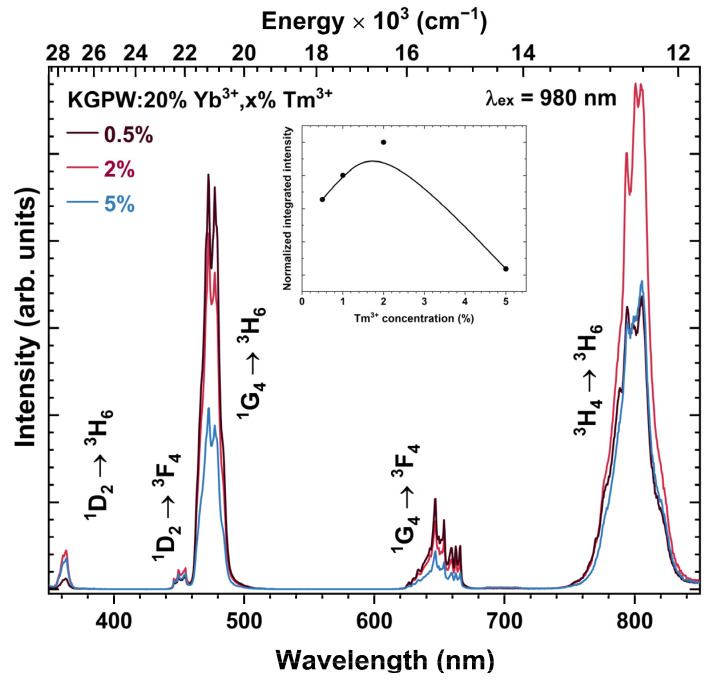
Upconversion emission spectra of KGPW: 20%Yb^3+^,Tm^3+^ as a function of Tm^3+^ concentration (λ_ex_ = 980 nm). The inset shows the Tm^3+^ concentration-dependent normalized integrated emission.

**Figure 6 materials-16-01305-f006:**
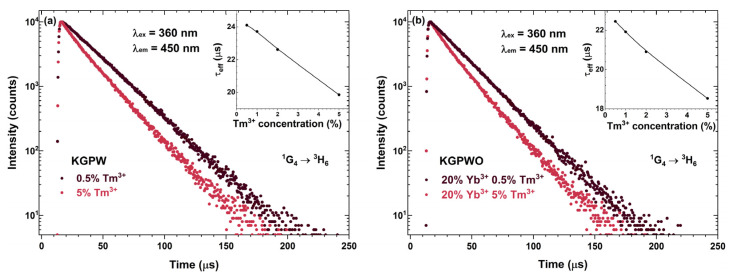
DC PL decay curves (λ_ex_ = 360 nm, λ_em_ = 450 nm) of KGPW: Tm^3+^ (**a**) and KGPW: 20%Yb^3+^,Tm^3+^ (**b**). Both insets show Tm^3+^ concentration-dependent *τ_eff_* values.

**Figure 7 materials-16-01305-f007:**
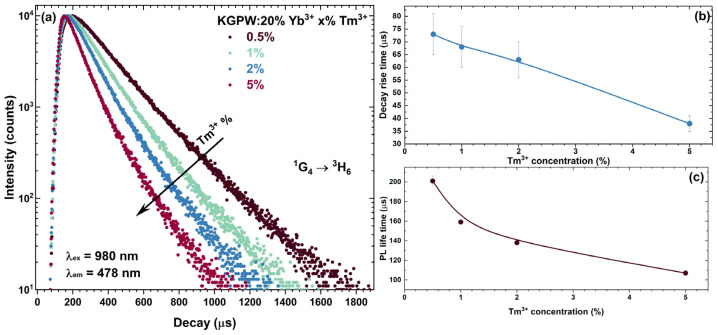
Tm^3+^ concentration-dependent: UC PL decay curves (λ_ex_ = 980 nm, λ_em_ = 478 nm) of: KGPW: 20%Yb^3+^,Tm^3+^ (**a**), PL rise time (**b**), and *τ_eff_* (**c**).

**Figure 8 materials-16-01305-f008:**
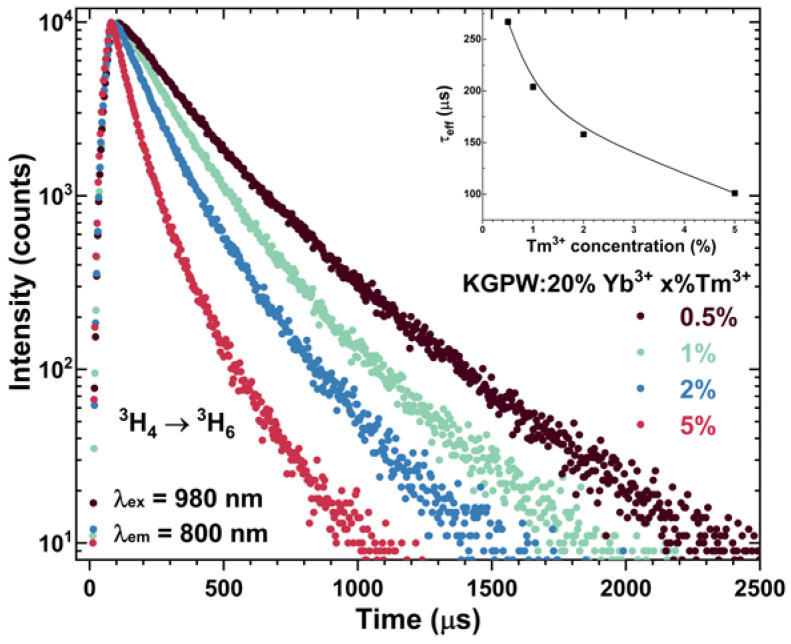
Tm^3+^ concentration-dependent UC PL decay curves (λ_ex_ = 980 nm, λ_em_ = 800 nm) of KGPW: 20%Yb^3+^,Tm^3+^. Tm^3+^ concentration-dependent *τ_eff_* values are given in the inset.

**Figure 9 materials-16-01305-f009:**
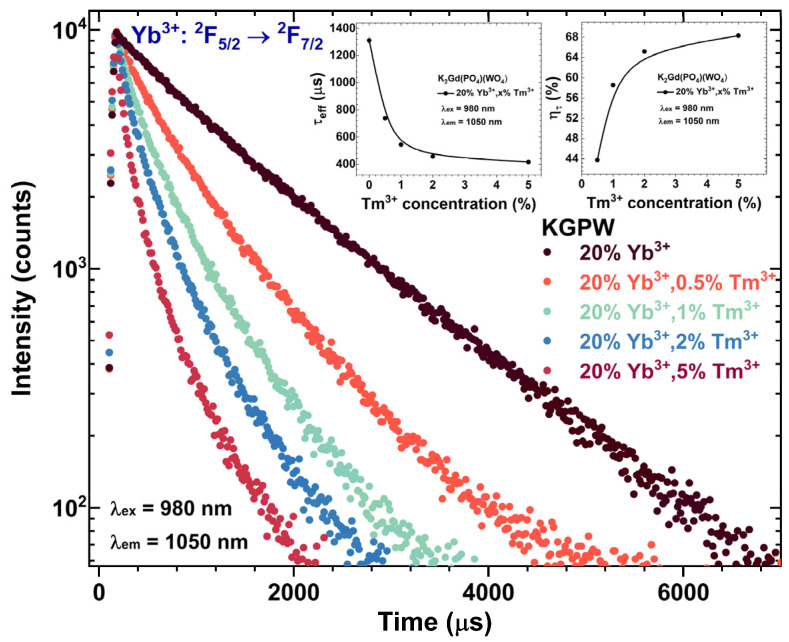
Tm^3+^ concentration-dependent PL decay curves of Yb^3+^ in KGPW: 20%Yb^3+^,Tm^3+^ compounds. The inset graphs show Tm^3+^ concentration-dependent *τ_eff_* and *η_tr_* values.

**Figure 10 materials-16-01305-f010:**
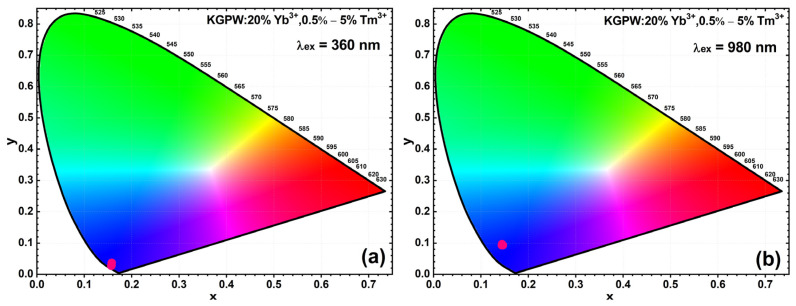
Tm^3+^ concentration-dependent color coordinates (in CIE 1931 color space) of KGPW: 20%Yb^3+^,Tm^3+^ samples excited with 360 nm radiation (**a**) and 980 nm wavelength laser (**b**).

## Data Availability

The data presented in this study are available on request from the corresponding author.

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
