# Peer review of "Optical Properties Investigation of Upconverting K2Gd(PO4)(WO4):20%Yb3+,Tm3+ Phosphors"

_materials, 2023, doi:10.3390/ma16031305_

Round 1
Reviewer 1 Report
The article is within the scope of the journal. The authors present the preparation and optical characterization of K2Gd(PO4)(WO4) doped with 20%Yb3+,Tm3+.
The design and methodology seem appropriate. Although a good bibliographic review is presented, it could be improved with more recent citations.
Regarding the identification of the crystalline phase, I understand that the authors should improve the text in paragraph lines 94-106. The authors do not present a refinement, for example by the rietveld method, that allows confirming the crystalline phase and the identified space group. Therefore, they should be careful in the way they report the structure. If the compounds are isostructural they should present the corresponding diagram or avoid phrases like "we can conclude that single-phase materials were obtained"
Reviewer 2 Report
This manuscript reported optical properties of upconverting K2Gd(PO4)(WO)4:20%Yb3+,Tm3+ phosphors. This manuscript is well organized, and fundamental upconverting properties have been investigated. I recommend minor revisions as follows.
1. What is the phonon energy of K2Gd(PO4)(WO)4?
2. What is the origin of the weak broad XRD peak in Fig.2?
More detail comments:
In this study, upconverting properties of K2Gd(PO4)(WO4) doped with Yb and Tm have been investigated. To the best of reviewer knowledge, there is no report on the upconverting properties of K2Gd(PO4)(WO4) host. As shown in the decay time, decay time decreased by increasing the concentration of Tm3+; therefore, up conversion luminescence should be emitted from the K2Gd(PO4)(WO4) doped with Yb and Tm. Furthermore, fundamental luminescence properties of upconversion luminescence spectra, decay time, energy transfer rate have been well evaluated. However, there are two points for revisions. Low phonon energy is important for unconversion luminescence as the author said. But, there is no information about that. Further, a broad XRD peak possibly due to amorphous phase was observed, so the reviewer needs to check whether the origin was from sample or glass substrate. Upconversion luminescence is common luminescence phenomenon; so the originality of the work is not so high, but the manuscript is well written, so the reviewer recommends minor revisions.
Reviewer 3 Report
Grigorjevaite et al. presents the investigation of the optical properties of K2Gd(PO4)(WO4):20%Yb3+,Tm3+ phosphors. There are some comments for the authors.
Major comments:
1. What is the reason that the author pick the concentration range of Tm3+ from 0 to 5%? What will happen if the concentration of Tm3+ goes higher? This should be mentioned in the manuscript.
2. Some of the figures miss 1%, like Figure 9. Please add it. Also, why did authors pick 0.5, 1, and 2? It looks very arbitrary. The data for 3% and 4% is highly recommended. If not, the authors should mention the reason why they are missing.
3. The authors claims that 2% has the highest intensity. If this is true, we would expect the PL lifetime also to be the shortest. But it seems that this is not shown in the lifetime measurement. The authors should explain this part.
Minor comments:
1. Line 27, replace “up-conversion” by “upconversion”. If the authors prefer to use UC to represent upconversion, they should UC rather than upconversion in rest of the manuscript.
2. Line 132 and 135, the symbols and numbers of excitation and emission are wrong.
3. The scale bar for Figure 10 is needed.
Round 2
Reviewer 3 Report
The manuscript is ready to be published.